# Aerosol Inhalation of Gene Delivery Therapy for Pulmonary Diseases

**DOI:** 10.3390/biom14080904

**Published:** 2024-07-25

**Authors:** Yiheng Huang, Jiahao Zhang, Xiaofeng Wang, Hui Jing, Hecheng Li

**Affiliations:** Department of Thoracic Surgery, Ruijin Hospital, Shanghai Jiao Tong University School of Medicine, Shanghai 200025, China; 120760910786@sjtu.edu.cn (Y.H.); zhangjiahao135@126.com (J.Z.); yexing669@foxmail.com (X.W.)

**Keywords:** gene delivery, aerosol inhalation, pulmonary disease, liposomes, polymers

## Abstract

Gene delivery therapy has emerged as a popular approach for the treatment of various diseases. However, it still poses the challenges of accumulation in target sites and reducing off-target effects. Aerosol gene delivery for the treatment of pulmonary diseases has the advantages of high lung accumulation, specific targeting and fewer systemic side effects. However, the key challenge is selecting the appropriate formulation for aerosol gene delivery that can overcome physiological barriers. There are numerous existing gene carriers under study, including viral vectors and non-viral vectors. With the development of biomaterials, more biocompatible substances have applied gene delivery via inhalation. Furthermore, many types of genes can be delivered through aerosol inhalation, such as DNA, mRNA, siRNA and CRISPR/Cas9. Aerosol delivery of different types of genes has proven to be efficient in the treatment of many diseases such as SARS-CoV-2, cystic fibrosis and lung cancer. In this paper, we provide a comprehensive review of the ongoing research on aerosol gene delivery therapy, including the basic respiratory system, different types of gene carriers, different types of carried genes and clinical applications.

## 1. Introduction

Recently, gene therapy has gained significant attention for potentially treating various diseases, including genetic disorders [1,2] (e.g., MRT5005 for treating cystic fibrosis [3]), neurological diseases [4] (e.g., Zolgensma for treating spinal muscular atrophy (SMA) [5]), cancer [6,7] (e.g., inhalation of IL-12 treating lung cancer [8]), and so on. In the past 20 years, many gene therapy drugs have already been approved for the treatment of several systemic diseases. For genetic diseases, Zynteglo [9] was approved to treat β-thalassemia, and Onpattro [10], from Alnylam Pharmaceuticals (Cambridge, MA, USA), was the first RNAi drug approved to treat amyloidosis in 2018. Some gene drugs have also been applied for cancer treatment, such as Yescarta [11], based on CD19-targeted chimeric antigen receptor (CAR) T cell immunotherapies, to treat large B cell-lymphoma; Gendicine to treat head and neck cancer; and Imlygic [12] as an oncolytic viral therapy to treat melanoma. The potential of gene therapy paves the way for precision medicine through the expression of specific genes. Since the COVID-19 pandemic, a large number of DNA and in vitro-transcribed messenger RNA (IVT-mRNA) vaccines have emerged [2]. In comparison with traditional vaccines, gene therapy offers distinct advantages, including easy production and the ability to encode a wide range of proteins [13]. Several DNA or mRNA vaccines have been approved for clinical use, such as the 1273-mRNA vaccine from Moderna [14,15] and the 162b2-mRNA vaccine from BioNTech (An der Goldgrube 12, Mainz, Germany). Moreover, gene therapy shows promise in cancer treatment by leveraging immune response activation [6]. Nonetheless, a significant challenge in the development of gene therapy lies in the delivery of genes, which requires suitable gene carriers [16].

Compared to traditional drug administration methods, aerosol inhalation of pulmonary drug delivery has the advantages of strong and fast efficacy, fewer systemic side effects, a lower drug dosage and repeatable administration [17]. Furthermore, the mucosal immune reaction plays an important role in the host’s response to SARS-CoV-2 infection. Consequently, direct pulmonary delivery can significantly trigger mucosal immunity, which results in better potency of the vaccine compared to traditional administration [18]. There are several methods of delivering pulmonary aerosols, such as direct inhalation, intratracheal and intranasal delivery [19]. A common concern with these methods is ensuring the safety of pulmonary delivery, including the prevention of pharyngeal edema, bronchial spasms or acute adverse reactions. There are many preclinical trials administering pulmonary delivery, but it is essential to consider the anatomical differences between mice and humans. There are four main kinds of inhalation instruments [20]: pressurized metered dose inhalers (pMDIs), dry powder inhalers (DPIs), nebulizers and soft mist inhalers (SMIs). Until now, the DPI [21] has been one of the most mature inhalation methods with the lowest error ratio and better stability compared to liquid aerosols.

Gene delivery has gained increasing attention as an approach to treat a wide range of diseases, including congenital genetic disorders, infectious diseases, cancer and chronic illnesses. Some medications have already advanced to clinical trials. For instance, in 2013, a phase IIb randomized, double-blind, placebo-controlled clinical trial was conducted to evaluate the aerosol inhalation of the pGM169/GL67A gene–liposome complex for the treatment of cystic fibrosis [22]. The results indicated a significant improvement of forced expiratory volume in 1s (FEV_1_) in the pGM169/GL67A group versus the placebo group after 12 months of treatment (3.7%, 95% CI 0.1~7.3; *p* = 0.046). Additionally, clinical trials are underway for aerosolized siRNA drugs, such as ARO-HIF2 for lung cancer from Arrowhead Pharmaceuticals [23] and QR-010 [24] for cystic fibrosis from ProQR Therapeutics, both of which are undergoing clinical trials. Nevertheless, for effective gene delivery, we still need to address challenges such as serum clearance [25], endosomal uptake [26] and off-target effects [27]. Researchers have dedicated substantial efforts to improving gene delivery. Despite extensive research into gene therapy [28] and several drugs having been approved by the FDA for treatment, aerosol gene therapy has not yet received approval for lung-related diseases [29].

Aerosol gene delivery to treat lung diseases seems to be a promising method due to its non-invasive nature and accurate targeting. However, the overall efficiency of pulmonary inhalation depends on various factors, including aerosol properties, particle characteristics [30], physiochemical properties and the physical attributes of the lungs [19].

In this review, we will elaborate on the current status of aerosol gene therapy, covering physiological aspects, various gene carriers, different types of carried genes and the diseases in which it can be applied.

## 2. Basic Characteristics of Inhalation Delivery

The most prominent advantage of inhalation administration is its non-invasive nature. As depicted in Figure 1, after inhalation of the formulation, it can traverse the airway and reach the deep alveolar space. However, achieving successful pulmonary gene delivery remains a challenging task. The primary obstacle is the intricate structure of the human respiratory tract, which requires a specific aerodynamic particle size. The size of particles plays an important role in the ultimate distribution [31]. Particles larger than 5 µm are more likely to accumulate in the upper respiratory tract, i.e., the oropharynx [32]. This is due to the collision of particles and finally results in deposition in the gastrointestinal tract. Droplets smaller than 1 µm are mostly exhaled [33]. Therefore, the optimal droplet diameter falls within the range of 1 to 5 µm, allowing for deposition in lower airways, terminal bronchioles and deep alveoli, which is facilitated by gravitational sedimentation.

Moreover, the formulation carrying the gene should be able to permeate a natural barrier, such as the mucus present on the airway epithelium [34]. This mucus is negatively charged, which may lead to the aggregation of cationic carriers and hinder the penetration of drugs. Additionally, surfactants on the alveoli containing phospholipids are also a significant challenge to effective delivery. Another obstacle is the mucociliary clearance mechanism of ciliated epithelial cells, which may exclude drugs or even direct them into the gastrointestinal tract. Mucosal clearance is a complex process involving the synergistic action of several components [35]. Macrophages also participate in phagocytosing inhaled drugs [30]. What is more, the alveoli fluid comprising alveoli surfactants and some immunoglobulins such as opsonin and other enzymes build up a self-defense mechanism against external invasions [19]. In addition, maintaining the stability of spheroids under hydrodynamic shear forces during nebulization is an important issue in gene delivery.

The integrity of genes may be compromised during nebulization, which means that the inclusion of protective materials such as polymers, liposomes, etc., is essential. Another factor influencing the effectiveness of the inhalation effect is the length of the gene. Studies have shown that a smaller plasmid is less sensitive to shear forces [36]. Some researchers have designed a novel approach using a fluorescence-labeled cross-linked cyclodextrin metal–organic skeleton material (CL-CD-MOF) as a model for inhaled particles. A fluorescence microscopic optical sectioning tomography system (f-MOST) [37] was employed to examine the entire lung’s structure. This innovative method allowed for three-dimensional, in situ and single-particle-level observation of the spatial distribution of CL-CD-MOF particles within the lung. It provided the highest resolution map of the entire lung distribution, marking a significant advancement in this field. To develop better treatment for pulmonary diseases, target genes need to be delivered effectively into target cells. Therefore, developing a suitable delivery system is an important issue. We will discuss this in the following sections, including viral vectors and non-viral vectors (Table 1).

## 3. Viral Vector-Based Carriers

Although a viral vector has the characteristics of high immunogenicity, difficult production and instability, it is still widely applied in gene delivery due to its high transfection efficiency [38]. There are various viral vectors used for gene therapies that are currently undergoing clinical trials, such as retrovirus vectors, adenovirus vectors and adeno-associated virus vectors (AAVs) [39]. What is more, some specific vectors such as herpes simplex viruses (HSVs), coxsackie virus and Newcastle disease viruses are also under research in clinical trials [40]. For example, IMLYGIC [41] is a type of HSV1 that can express the immunostimulatory protein GM-CSF (granulocyte–macrophage colony-stimulating factor), thereby exerting a therapeutic effect on melanoma. It was approved by the FDA for the treatment of melanoma in 2015.

### 3.1. Lentivirus

A lentivirus is a specific type of retrovirus with distinct characteristics such as low immunogenicity and stable expression. For example, the aerosol inhalation of lentiviral-encoded carboxyl-terminal modulator protein (CTMP) is proven to suppress lung tumor growth and increase apoptotic proteins [42]. A lentivirus is also applied in treating CF. Cooney et al. [43] utilized a feline immunodeficiency virus-based (FIV-based) lentiviral vector to carry CFTR (FIV-CFTR) to treat CF pigs through inhalation. After two weeks of treatment, the tracheal airway surface liquid pH and cAMP-stimulated current were increased, which indicated that FIV-CFTR could improve the anion channel defect. The duration of gene expression is another key point in gene delivery therapy. Martin et al. [44] adopted the repeated administration of a VSV-G pseudo-typed HIV-vector-carrying luciferase report gene to intervene in C57Bl/6 mice and observed bioluminescence over 12 months. The results showed that early consecutive administration with shorter intervals was more effective in gene expression. However, as a type of retrovirus, a lentivirus is likely to integrate into host cells randomly and cause unstable expression [45], which needs further exploration in the future.

### 3.2. Adeno-Associated Virus Vector (AAV)

AAV is a widely used viral vector for aerosol inhalation. Some researchers administered intratracheal AAV1 to carry the sarcoplasmic reticulum Ca^2+^-ATPase 2a (SERCA2a) gene to treat pulmonary hypertension (HP). The results showed that the administration of AAV1-SERCA2a can reduce pulmonary artery pressure and vascular resistance effectively in animal models [46]. AAV2 vectors carrying cystic fibrosis transmembrane conductance regulator (CFTR) complementary DNA (cDNA) have also been employed for cystic fibrosis [47]. In a multicenter, double-blind, phase II clinical trial, CF patients with mild lung disease were enrolled. The results showed that repeated aerosol inhalation of AAV2-CFTR exhibited good biocompatibility and the improvement in FEV1 (forced expiratory volume in one second) (*p* = 0.04) compared to the placebo group after 30 days. Additionally, Macloughlin et al. [48] discovered that aerosolized AAV2/6 vectors carrying human Iκ-Bα, which is the inhibitor of NF-κB, can significantly attenuate endotoxin-mediated lung injury. This finding offers a potential therapy for acute lung injury.

### 3.3. Adenovirus

An adenovirus is a broadly applied vector in vaccines. Some researchers reported that the aerosol delivery of an adenoviral-based COVID-19 vaccine following two inactivated vaccines results in higher levels of neutralizing antibodies (Nabs). A comparison between inhaled Ad5 COVID-19 (I-I-Ad5) and intramuscularly administered inactivated vaccines revealed that the I-I-Ad5 group had a better therapeutic effect [49]. While adenovirus vectors have shown promise in gene delivery and vaccination, it is worth noting that the immune response triggered by the AAV remains a challenge [50]. Adenoviruses that induce an immune response may potentially hinder the therapeutic effect of gene delivery.

## 4. Non-Viral Vectors

### 4.1. Liposomes

Liposomes are widely used drug carriers whether for hydrophobic or hydrophilic genes [51]. They have the advantages of easy preparation, good biocompatibility and minimal toxicity [52]. Cationic liposomes have been proven to be efficient in delivering genes, owing to their cationic lipids capable of encapsulating negatively charged nucleic acids [51,53]. Their positively charged nature facilitates cellular uptake. Conversely, non-cationic liposomes tend to possess superior immune-stimulating properties [51]. For instance, a niosome, which is a single-chain surfactant, is a more stable and easily producible alternative to a traditional liposome [54]. Wang et al. [55] designed a liposome-encapsulated plasmid encoding influenza A/PR/8/34 (H1N1) hemagglutinin (pCI-HA10). This liposome encapsulated pCI-HA10 and triggered robust mucosal, cellular and humoral immune reactions in mice after intranasal (i.n.) delivery compared to intramuscular (i.m.) delivery. Notably, there was a significant increase in IgG, IgA and T cells in the bronchoalveolar lavage (BAL) fluid. However, the following potential drawback needs to be considered: it may interact with negative proteins, which may result in immune clearance or toxicity [53]. Therefore, the modification of the surface of liposomes needs to be taken into consideration.

The commonly employed components for modification include phosphatidylcholine (PC), dioleoyl phosphatidylethanolamine (DOPE) and cholesterol (Chol) [33].

Studies have shown that the incorporation of dioleoyl phosphatidyl-ethanolamine (DOPE) into liposomes offers a dual benefit by improving cellular uptake while reducing cytotoxicity [56]. Furthermore, it has been observed that there is improvement in the stability of DNA during nebulization. Successful transfection of plasmid DNA (pDNA) was achieved after 21 days of jet nebulization when it was encapsulated in a mixture of DOPE linked to DOTMA (N-[1-(2,3-dioleyloxy)propyl]-N,N,N- triethylammonium) [57]. Davies et al. [58] designed pDNA (5.6 kb)/GL67A, which consisted of Genzyme Lipid 67, DOPE and DMPE-PEG5000 at a ratio of 1:2:0.05. These complexes were subjected to testing through a next-generation pharmaceutical impactor (NGI) for 4 min at a flow rate of 15 L/min (±5%) through jet, mesh and ultrasonic nebulizers. The results demonstrated the complete protection of the DNA during jet nebulization. In a double-blind, phase 2b clinical trial, a pGM169/GL67A gene–liposome complex was nebulized every 28 days for a year in patients with cystic fibrosis [22]. A significant effect was observed in the pGM169/GL67A group versus the placebo after 12 months (3.7%, 95% CI 0.1–7.3; *p* = 0.046). Recent studies have indicated that the addition of compounds such as 1,2-dioleoyl-3-trimethylammonium-propane (DOTAP), DOTMA and 2,3-dioleyloxy-N-[2-(sperminecarboxamido)ethyl]-N,N-dimethyl-1-propanaminium (DOSPA) can also enhance the transfection efficiency in the lungs. Rosada et al. [59] combined egg phosphatidylcholine (EPC), DOPE and DOTAP to create liposomes carrying pDNA encoding HSP65 as a tuberculosis (TB) vaccine. The DNA–hsp65–liposome complex exhibited increased IFN-γ and preservation of lung parenchyma in a mouse TB model. Furthermore, Allon et al. found that increasing the concentration of phospholipids (PSs) can facilitate the escape of the gene from endosomes [60]. The results indicated that, after the addition of PSs to liposomes, cellular uptake increases while immunogenicity decreases. Another study found that the inclusion of surfactants can improve the transfection efficacy of DNA. The limiting factor is the toxicity of geminin surfactants that comprise two charged heads and an alkyl tail.

Dahlman et al. [61] conducted a study focused on screening cluster-based lipid-nanoparticles (LNPs). These LNPs were generated through the microfluidic method and utilized a previously optimized oligomer–lipid conjugate known as 7C1 to carry luciferase-encoding mRNA. They analyzed the effects of different chemical compositions and molar ratios of the following nanoparticle components: major lipids, neutral or cationic auxiliary lipids, cholesterol, 7C1 and polyethylene glycol (PEG). The results indicated that the efficiency of lung delivery could be enhanced by a higher PEG ratio. Maintaining a suitable ratio of PEG and non-helper lipids is also crucial for the stability of LNPs [62]. Rachel Yoon Kyung Chang et al. designed LNPs composed of 7C1, cholesterol, PEG and DOTAP. The new LNPs showed better stability and enhanced accumulation in the lungs. Lokugamage et al. [61] also conducted experiments to assess the ratio of PEG and the helper lipids of LNPs. They designed a nanoparticle referred to as ‘nebulized lung delivery 1 (NLD1)’, which features a high PEG content and cationic lipids. NLD1 was found to improve mRNA delivery to the lungs.

Several studies have suggested that the addition of PEG can increase the stability of LNPs, thereby preventing their aggregation and improving endosomal escape [63]. However, it is worth noting that an excessive amount of PEG on the surface of LNPs may inhibit cellular uptake, thus reducing their transfection efficiency. In addition to adding PEG, Kim et al. screened a small library of LNPs and identified a particular type of LNP (nLNP) containing a cholesterol analog, β-sitosterol, which can improve the endosomal escape of LNPs. The results indicated the better diffusion of nLNP across the mucus barrier. The delivered gene cystic fibrosis transmembrane conductance regulator (CFTR) mRNA was highly expressed after nebulization in cystic fibrosis pigs.

As for the gene-editing tool delivery system, adeno-associated virus (AAV) remains a primary method. However, it faces challenges such as long-term expression and potential off-target effects. Li et al. [64] synthesized and screened a combinatorial library of biodegradable ionizable lipids to build an inhaled LNP for delivering either mRNA or CRISPR-Cas9 gene editors. They created a three-component reaction (3CR) system in which the nitro-ricinoleate acrylate (NRA) linker was coupled to a fatty alcohol (lipid tail) and a head group containing primary, secondary or tertiary amines. The RCB-4-8 LNP was selected as it is biodegradable and suitable for repeatable inhalation (Figure 2). The results showed that the RCB-4-8 LNP can increase the concentration of luc-mRNA 100-fold more than the MC3-LNP from Alnylam, a product on the market since 2018. This article presents a novel approach involving high-throughput screening to improve the delivery of ionizable LNPs, which has the potential to broaden the application of CRISPR-Cas9 gene editing and mRNA therapy.

In conclusion, the inhalation of liposomes carrying genes has proven to be effective in both humoral and mucosal immune responses. However, it is important to address the related challenges, including the stability of liposomes during nebulization and aqueous dispersions. More research effort should be directed towards storage and the enhancement of liposome stability in these contexts.

### 4.2. Polymers

#### 4.2.1. Poly-Ethylenimine (PEI)

Polymeric nanoparticles include poly-ethylenimine (PEI), poly(lactic-co-glycolic acid) (PLGA) and polylactic acid (PLA) [66]. Among those polymers, poly-ethylenimine (PEI) stands out as a typical cationic polymer that is widely used in gene delivery due to its electrostatic interaction with genes. PEI is typically classified into the following two forms: linear PEI (l-PEI) and branched PEI (b-PEI). The transfection efficiency is often related to the structure and molecular weight of PEI [67]. A higher molecular weight leads to better transfection but also results in increased cytotoxicity. Therefore, it is important to select a proper PEI that can balance the transfection efficiency and biological safety. Shim et al. [68] employed PEI to carry SARS-DNA encoding spike proteins (pci-S) intranasally (i.n.) to mice. Notably, the mucosal IgA and IgG were much higher in the PEI/PCI-s (i.n.) group compared to the group receiving pci-S alone. This enhancement in immune response can be attributed to the stability of the PEI–DNA complex. In another study, Jia et al. [69] utilized PEI to carry IL-12 DNA to treat osteosarcoma (OS) lung metastases in a nude mouse model. The results showed that, following two weeks of aerosol administration, both subunits of IL-12 were highly expressed in the lungs, which resulted in higher anti-tumor efficacy in the PEI-IL-12 aerosol group. What is more, PEI-IL-12 exhibited good targeting to the lung, with high IL-12 expression in the lung within 24 h but no expression observed in the liver.

As for traditional nebulizers, there is a problem with high shear forces. SAWs (surface acoustic waves) have been used to nebulize formulations carrying siRNA into 1–5 µm particles, which can achieve deep lung accumulation [70]. Researchers have tested the stability of PEI/siRNA in different N/P proportions of 30, 60 and 90. The results showed that the size and potential of PEI/siRNA remained the same after nebulization in all N/P proportions, which indicated good stability [70]. What is more, A549 lung carcinoma cells expressing luciferase were knocked down after being transfected with nebulized PEI/siRNA.

There are also various approaches available for modifying PEI to form a more stable formulation. Research has demonstrated that the stability of the PEI/DNA complex surpasses that of other complexes such as DOPE/DNA or cholesterol–DOPE/DNA. Moreover, the specific structure of PEI also plays an important role in gene delivery. Different types of PEIs, including branched PEI (b-PEI), linear PEI (l-PEI) and pegylated PEI (peg-PEI), when combined with DNA, have all been shown to protect the integrity of DNA subjected to jet or ultrasonic nebulizers [71].

In some investigations, PEI has been employed as a carrier to encapsulate Wilms’ tumor gene 1 (WT1-siRNA) for the treatment of melanoma lung cancer metastasis [72]. WT1 is known to be related to the progression of multiple tumors, including lung cancer. The results indicated that, when mice with B16-F10 melanoma lung metastases were administered 25 ug gene–PEI complexes via inhalation, there was a notable decrease in newly formed vessels in tumor sites and an increase in tumor apoptosis.

Although PEI is a well-established delivery system, it has some drawbacks, such as cytotoxicity [73], mainly caused by its cationic charge density [74]. To address this issue, some researchers have dedicated their efforts to modifying the PEI to reduce toxicity. One common strategy involves PEGylation, as well as modifications of natural components like HA or chitosan. PEG is a biodegradable hydrophilic polymer and currently the most commonly used material for improving the biocompatibility of PEI. It can improve the biocompatibility of the gene complex through shielding the positive charge on the surface of the carriers [75]. In addition, there are lots of hydroxyl groups on PEG that can form a hydration layer on the surface of the carrier. This will help prevent aggregation and thus avoid phagocytosis of the reticuloendothelial (RES) system. Kleeman et al. utilized a poly-ethylenimine-g-poly(ethylene glycol) copolymer (PEG-PEI) to deliver plasmid DNA (pDNA), which can maintain a spherical structure after nebulization [76].

#### 4.2.2. Other Polyplexes

In addition to PEI, various other materials have been utilized for gene delivery such as poly-amidoamine (PAMAM), polyacrylic acids (PAA), Poly-L-lysine (PLL) and other self-assembly nanoparticles [77].

Kesharwani et al. [78] created an upconversion nanoparticle (UCNP) with a surface coating of polyacrylic acids (PAAs). PAA serves as a linker between the nano-core and DNA AS1411, which acts as an aptamer-targeting tumor. Upon reaching the tumor sites, the MMP-2-cleavable peptide can be degraded, and thus releases siRNA outside. Mice with lung tumors exhibited enhanced tumor inhibition and prolonged survival after the nebulization of UCNP-siRNA-PAA-AS1411.

Moreover, poly(lactic-co-glycolic acid) (PLGA) is a widely used and successful polymer composed of two monomers, lactic acid and glycolic acid, which are known for their good biocompatibility [79]. For example, some researchers have employed PLGA-PEG as a siRNA carrier. While siRNA tends to accumulate predominantly in liver tissue, a small fraction can be delivered to other tissues. Bai et al. developed a novel nano-delivery system through the self-assembly of PLGA-PEG and a self-created cationic lipid molecule G0-C14 (PPGC) to form siIL11@PPGC, which carries siRNA targeting IL11 (siIL11@PPGC). IL-11 plays an important role in idiopathic pulmonary fibrosis (IPF), which can differentiate fibroblasts into collagen-secreting, actin alpha 2 and smooth muscle-positive (ACTA2^+^) myofibroblasts. The results indicated that the aerosol inhalation of siIL11@PPGC could alleviate fibroblast differentiation in mice with fibrosis. The pulmonary function was significantly improved, with increased compliance (Crs) and forced vital capacity (FVC) [80].

#### 4.2.3. Synthetic Esters

Recently, the incorporation of biodegradable components such as esters has emerged as a promising method to improve gene delivery efficiency. Suberi et al. [81] designed a biodegradable poly(amine-co-ester) (PACE) polyplex for inhaled spike protein mRNA delivery as a mucosal vaccine for the treatment of SARS-CoV-2. PACE-mRNA exhibited robust transfection in both epithelial and antigen-presenting cells (APCs). The results indicated that intranasal vaccination of spike protein mRNA via this formulation could induce cellular and humoral adaptive immunity, leading to the accumulation of CD8+ T cells. Furthermore, PACE-mRNA can also prevent mortality in K18-Hace2 mice during exposure to viruses. Ma et al. [65] created a star polymer composed of PDMAEMA-POEGMA (poly-dimethylaminoethyl methacrylate poly-oligo (ethylene glycol) methyl ether methacrylate) to carry siRNA targeting βIII-tubulin (TUBB3) and Polo-Like Kinase 1 (PLK1) mRNA (Figure 2 is a schematic of the in vitro and in vivo function of star-siRNA nanoparticles). The results demonstrated that the nebulization of the siRNA-star nanoparticles can successfully deliver cargo to the target cytosol, leading to delayed tumor growth in orthotopically implanted immune-deficient mice.

Patel et al. [82] designed hyperbranched poly(beta amino esters) (hPBAEs) to carry mRNA. In vivo results indicated a substantial presence of luciferase protein in all five lobes of the lungs within 24 h of inhalation. Rotolo et al. conducted a screening of 166 polymeric nanoparticle formulations and identified P76, a poly-β-amino-thio-ester polymer (PBATE) [83]. They found that p76 could deliver various types of mRNA to the lungs while maintaining a high level of safety and tolerance in the meantime. The results showed that inhalation of p76/CRISPR-Cas13a mRNA achieved similar efficacy in SARS-COV-2-infected mice, but with only a fraction (1/20) of the antibodies.

### 4.3. Peptide-Based Nanoparticles

In addition to traditional endocytosis of nanoparticles into cells, peptide-modified carriers exhibit promising transport and targeting capabilities [84]. Positively charged peptides can bind to nucleotides through charge interactions. Compared to viral vectors, peptide vectors exhibit lower cytotoxicity, lower immunogenicity and higher biodegradability. One commonly used peptide is the cell-penetrating peptide (CPP), typically composed of 5–30 amino acids, which can improve endocytosis [85]. The charge, molecular weight and size of CPPs may influence the transfection efficiency [85]. CPPs are usually coupled with plasmid DNA or siRNA. An example of a CPP is the trans-activator of transcription (TAT), derived from HIV (RKKRRQRRR). Kawabata et al. dimerized the TAT peptide (HIV1 trans-activator) to formulate dTAT-NP as a carrier to deliver angiotensin II type 2 receptor (AT2R) plasmid DNA [86]. The in vivo results indicated that the bolus intratracheal administration of 0.7 µg of VEGF-DNA via dTAT four times can significantly trigger tumor cell apoptosis in LLC-inoculated mice. Ishiguro et al. also utilized a TAT peptide to modify nanoparticles for AT2R gene delivery [87]. They created a dimerized TAT (dTAT) and plasmid DNA (pDNA) complex through calcium chloride (CaCl_2_), forming dTAT-pAT2R-Ca^2+^. Their findings demonstrated that dTAT-pAT2R-Ca^2+^ could inhibit tumor growth more effectively through intratracheal (I.T.) rather than intravenous (I.V.) delivery in Lewis lung carcinoma allografts. Poloxamine has been employed as a polymer for gene delivery, but the transfection efficiency lags behind that of viral vectors. Guan et al. [88] created a peptide–poloxamine 704 (T704) nanoparticle termed an mRNA-based complex (mRNA-TC) through self-assembly to treat cystic fibrosis (CF). The peptide is composed of the following three components: an anchor moiety to interact with T704, a cationic moiety and a target moiety. This peptide-based complex can significantly increase the accumulation of Sleeping Beauty (SB) Transposon pDNA and SB100X-mRNA. As a result, the CFTR level remained consistently high in CF mice.

### 4.4. Natural Biocompatible Components

#### 4.4.1. Chitosan

Chitosan is one of the most common biodegradable natural materials for gene delivery [89,90]. Chitosan possesses the characteristics of biocompatibility and a mucus-adhesive ability [91,92]. Raghuwanshi et al. [93] utilized biotinylated chitosan to deliver plasmid DNA (pDNA) encoding the nucleocapsid (N) protein of SARS-CoV-2. Additionally, they modified the chitosan with an anti-DEC-205 single-chain antibody (scFv) to facilitate the targeting of dendritic cells (DCs). The intranasal administration led to increased levels of IgG and IgA against N-protein. Modification of PEI with biocompatible components is another method. A common method is adding chitosan into PEI. Nascimento et al. designed a chitosan nanoparticle for the delivery of the mitotic checkpoint siRNA Mad2 to treat NSCLC. The results indicated that nanoparticles can also target EGFR through the modification of chitosan [94].

Chitosan can also be combined with other natural components to enhance the stability and transfection efficiency. Some researchers combined chitosan (CS), Arg and TCEP (reducing agent—tris(2-carboxyethyl) phosphine) to form a CS-Arg/TCEP (CAT) nanogel (Figure 3) [95]. The CAT nanogel exhibits good biocompatibility and water solubility. Upon inhalation of the CAT nanogel, chitosan adheres to the mucus and releases TCEP through disulfide cleavage in aqueous solution. After the nebulization treatment of balb/c mice with the allergic asthma model, the mucus obstruction was significantly alleviated and the airway inflammation was improved.

Indeed, chitosan is a versatile material not only for delivering nucleic acids like siRNA and mRNA but also for gene-editing systems such as CRISPR-Cas9. Zhang et al. [96] conjugated poly (ethylene glycol) monomethyl ether (mPEG) to chitosan with chitosan of varying molecular weights. This newly formed PEGylated chitosan/pSpCas9-2A-GFP nanocomplex demonstrated excellent stability after the nebulization and digestion of DNase. It was shown to have good mucus penetration and efficient transfection capability. This article provides a new stage for combining PEGylation and chitosan together for delivering CRISPR-Cas9.

#### 4.4.2. Exosome-Based Vectors

Recently, exosomes have emerged as an effective vehicle for gene delivery [97]. For instance, some researchers harnessed lung exosomes to deliver target genes, which has the advantages of better therapeutic properties, drug retention, higher efficacy, evading immune clearance and more pulmonary accumulation [98]. They tested the transfection efficacy of the following three formulations: lung-exo, HEK-exo and lipid nanoparticles (LNPs). The results showed that mRNA delivered by lung-exo could penetrate alveolar epithelial cells and accumulate deep within the lungs, whereas the other formulations mainly retained in air tube epithelial cells.

Inhalable exosome-based gene delivery can also be utilized as an mRNA vaccine. The exosome derived from lung spheroid cells (LSCs) has the characteristics of homology targeting to the lung tissue, which can realize more efficient pulmonary delivery and accumulation [99]. LSCs are therapeutic lung cells extracted from transbronchial lung biopsies and cultivated in vitro through the 3D method. Moreover, exosomes can also be taken up by DCs, triggering an immune response as a vaccine. Wang et al. [100] designed a recombinant SARS-CoV-2 receptor-binding domain (RBD) conjugated to a lung spheroid cell (LSC)-derived exosome (LSC-Exo) to form RBD-Exo (as Figure 4 shows). After the inhalation of RBD-Exo in SARS-CoV-2-infected hamsters, there was a significant induction of antigen-specific IgA and T cell response in the lungs. In addition, RBD-Exo exhibited better lung delivery and good stability at room temperature for up to three months, which can greatly reduce vaccine costs.

In addition to exosomes, secretomes can also be harnessed for inhalation. Dinh et al. [101] utilized natural components from the secretome of lung spheroid cells (LSC-Sec). The inhalation of LSC-Sec can achieve lung repair in pulmonary fibrosis. The results of lung pathology sections showed that the inhalation of LSC-Sec could reduce fibrosis and facilitate vascular repair in silica or bleomycin-induced pulmonary fibrosis in rodents.

Biocompatible nanoparticles can be endowed with the smart response ability. Pan et al. [102] covered the lipid core with gelatin and glutamic acid to load Fc, forming FcNLC(F)@PC. The outer amino acid layer can be degraded by MMP2 and release the inner drug. The complex can trigger ferroptosis in adenocarcinoma-bearing mice through inhalation. Zhang et al. [103] developed an endogenous recombinant ribosomal protein to deliver mMMP13 and KGF (mMMP13@RP/P-KGF). After the inhalation of mMMP13@RP/P-KGF, the formulation is largely deposited in pulmonary alveoli. It can also realize stepwise release by releasing the outer KGF first, followed by the release of MMP13 after the recombinant ribosomal protein is degraded. The KGF can promote the proliferation of epithelium cells and MMP13 can accelerate the degradation of ECM, thus alleviating IPF.

## 5. Different Types of Genes for Aerosol Inhalation

### 5.1. DNA

Aerosol inhalation of DNA is a common gene delivery method. It can be applied in the treatment of many diseases, such as cystic fibrosis (CF), chronic obstructive pulmonary disease (COPD) and lung cancer. Aerosol inhalation of a recombinant adenovirus carrying cystic fibrosis trans-membrane regulator (CFTR) (Ad2/CFTR) DNA was proven to be safe and feasible [104]. However, the duration of the gene expression and the transfection efficacy in airway epithelium needs further improvement. Recently, a phase IIa/b clinical trial [58] evaluated the safety and feasibility of using the cationic liposome GL67A to deliver pDNA (pDNA/GL67A) to CF patients. The results showed that pDNA/GL67A could retain its biological function after nebulization and the CFTR gene expression was high.

### 5.2. mRNA

Compared to DNA delivery, the aerosol inhalation of RNA has several advantages. RNA can be translated into proteins directly without entering the nucleus. The transfection efficiency is higher and the immune response is lower. At the same time, RNA also faces the challenge of degradation in vivo. Aerosol inhalation of mRNA can be applied in many diseases, such as respiratory diseases, viral infections like COVID-19, genetic diseases and lung cancer. For example, the direct inhalation of cancer suppressor genes has been a common approach in cancer therapy. Some researchers have utilized the delivery of genes like IL-12 [69] and p53 DNA [105] for treating lung metastasis. In recent years, chemically modified messenger RNA (modRNA) has attracted a lot of research for its low immunogenicity and high transfection efficiency [106]. Haque et al. [107] utilized chitosan-coated PLGA nanoparticles (RG 752 H NPs) to deliver CFTR modRNA (RG 752 H/CFTR) to treat cystic fibrosis. The results showed that, after the nebulization of RG 752 H/CFTR in CFTR^−/−^ mice, chloride decreased significantly and FEV1 improved. With the outbreak of SARS-CoV-2, inhaled mRNA vaccines have gained significant attention and research interest, and exhibit great potential [108].

### 5.3. Other Types of RNA

Along with aerosol inhalation of mRNA to realize gene expression, the inhibition of gene expression is another RNA therapeutic approach [109]. For example, small interfering RNA (siRNA), microRNA (miRNA) and antisense oligonucleotide (ASO) are common types of RNA used to inhibit gene expression, Among them, siRNA is one of the most common involved in gene therapy methods because it allows for the specific targeting and precise regulation of genes [110]. It can be applied in various diseases, such as genetic diseases, neurological disorders and cancer. For example, the inhalation of Bcl2-siRNA with PEI conjugating to the chemo drug DOX (as Figure 5 shows) has been shown to significantly enhance the anti-tumor efficacy of mono-chemotherapy [111]. Another polymer nanoparticle system (PEI-HZ-DOX/Bcl2-siRNA) is formed through the electrostatic interaction between cationic PEI-HZ-DOX and anionic Bcl2-siRNA. This novel PEI-HZ-DOX/Bcl2-siRNA exhibited an enhanced rate of apoptosis against a B16F10 metastatic lung cancer model, with a significant accumulation in lung tissue [112].

### 5.4. Clustered Regulatory Interspaced Short Palindromic Repeats (CRISPR)/CRISPR-Associated Protein 9 (CRISPR/Cas9)

CRISPR/Cas9 is a gene-editing technology with wide-ranging applications in various diseases [113]. Recently, researchers found that the inhalation of CRISPR/Cas9 allows for increased accumulation and a reduction in off-target effects. This breakthrough has potential in the treatment of pulmonary diseases such as lung cancer, cystic fibrosis and other chronic diseases [114]. Zhang et al. [96] successfully created a PEGylated chitosan/pSpCas9-2A-GFP nanocomplex that exhibited good intracellular uptake, effective mucus permeation and good stability when administered through nebulization. Their article demonstrated the feasibility of the inhalation of CRISPR/Cas9 as a delivery method. CRISPR/Cas9 can be inhaled in various ways, including as liquid and powder formulations. What is more, CRISPR/Cas9 can involve different gene types such as plasmid DNA (pDNA) (encoding Cas9 protein), mRNA (mRNA encoding Cas9 protein) and a Cas9 protein–sgRNA complex [115]. The inhalation type of CRISPR/Cas9 could be in liquid or powder form. Meanwhile, the aerosol inhalation of CRISPR/Cas9 still faces some challenges, such as finding suitable delivery systems, the persistence of off-target effects and the need for biocompatibility. Finding a suitable delivery formulation of CRISPR/Cas9 is a main obstacle in clinical translation and requires the payload of formulation, stability during nebulization and the protection of CRISPR/Cas9 from degradation. The characteristics of formulations such as solution properties, engineering of particles and devices for aerosol would impact the inhalation of CRISPR/Cas9. Although the delivery system of gene-editing tools has achieved great progress in many diseases, such as lung cancer, cystic fibrosis and asthma [116], the application of inhalation delivery in pulmonary diseases still has a long way to go.

## 6. Application of Aerosol Inhalation of Gene Delivery in Pulmonary Diseases

### 6.1. Vaccines

Since the SARS-CoV-2 outbreak, extensive research has been conducted on nucleic acid-based vaccine delivery through inhalation. In contrast to traditional inactivated or attenuated virus vaccines, nucleic acid-based vaccines can be basically categorized into DNA and mRNA vaccines [117]. The greatest advantage of the inhalation of vaccines is that it is non-invasive compared to traditional vaccines, which involve injection with needles. It also has the advantage of eliciting more precise humoral and cellular immune responses in the host [118]. Moreover, aerosol vaccines are easily designed and amenable to large-scale production. As for these two types of nucleic acid vaccines, DNA vaccines need to reach the nucleus of host cells, while mRNA vaccines can just reach the cytoplasm to function. Notably, several inhaled nucleic acid-based vaccines have already emerged. For example, aerosol inhalation of the adenovirus type-5 vector-based COVID-19 vaccine (Ad5-nCoV) has undergone a single-center, phase 1 clinical trial in China [108] to evaluate the safety and immunogenicity of the vaccine. The results showed that two doses of aerosol Ad5-nCoV could achieve the same immune response as intramuscular injection and had good safety. Aerosol inhalation in liquid form has high requirements for cold-chain storage. Ye et al. [119] developed a microcapsule to deliver the SARS-CoV-2 RBD antigen through the inhalation of dry powder and demonstrated effective protection against SARS-CoV-2 in mice. Aerosol gene delivery can be applied in other infectious diseases such as influenza and measles. For example, measles, caused by the measles virus, is highly contagious and has a high mortality. Some researchers conducted a clinical study in children randomized into an aerosol group and an injection group. They found that the aerosol measles vaccine group had superior immunogenicity to the subcutaneous injection group [120]. The results showed that the antibody level was much higher in the aerosol group (52–64%) than in the injection group (4–23%). Inhalation vaccines are also promising in influenza. A phase III clinical trial in children aged 6–59 months found that the influenza vaccine (Flumist) was more effective than a traditional inactivated vaccine [121]. What is more, the aerosol inhalation of vaccines can be applied to other airborne diseases. Inhaled nucleic acid-based vaccines also exhibit potential for other infectious and chronic diseases, such as HIV, tuberculosis [122] and lung cancer. One of the most significant advantages of inhalation of nucleic acid-based vaccines is the induction of a mucosal immune response compared to traditional administration [123]. However, it is important to acknowledge that the aerosol inhalation method still faces various challenges, including physiological barriers [124], mucosal clearance [123] and the development of efficient delivery vectors.

### 6.2. Cystic Fibrosis and Other Chronic Diseases

Cystic fibrosis (CF) is one of the most prevalent autosomal recessive diseases and is created by a mutation in the transmembrane conductance regulator (CFTR) [125]. Until now, the CF treatment has remained ineffective. The current objective in treating CF is attenuating the progression of lung damage. CF remains incurable in its current state. Many researchers have worked on developing the nebulization of genes for direct pulmonary delivery. For example, the intratracheal administration of a combination of fluorinated polymeric CXCR4 antagonist AMD3100 and siRNA can significantly reduce α-SMA in CF [126]. Given the mutation of CFTR in CF, various studies have focused on delivering the functional CFTR gene to alleviate CF. Kim et al. employed the nebulization of LNPs carrying CFTR mRNA, leading to good outcomes in cystic fibrosis pigs. Guan et al. [88] developed a peptide-based nanoparticle that could increase the CFTR level in CF mice. What is more, the use of an AAV2 vector carrying CFTR cDNA led to a great improvement in FEV1 in cystic fibrosis patients [47]. MRT5005, which encodes the CFTR protein as an inhaled mRNA drug, had undergone a randomized, double-blind, placebo-controlled phase 1/2 clinical study (NCT03375047) and received Fast Track designation for treatment of CF by the FDA [3]. The results showed that the inhalation of MRT5005 was well tolerated with no severe side effects, and the ppFEV1 (percent predicted forced expiratory volume in 1 s) was improved significantly in the middle-dose group (16 mg).

The inhalation of gene therapy can also be applied in other chronic diseases. Pulmonary hypertension (PH) is characterized by elevated pulmonary vascular resistance and a dysfunctional right ventricle (RV) [127]. Some studies showed that the sarcoplasmic reticulum Ca^2+^-ATPase 2a (SERCA2a) is downregulated in PH patients. In order to change this situation, Aguero et al. have designed an aerosolized inhalation of AAV1-based SERCA2a to treat PH, which exhibited good efficacy in animal models. In addition, aerosol gene delivery therapy can be delivered to realize systemic effects. Currently, the treatment of chronic diseases usually requires repeated injections or oral administration, which leads to poor patient compliance and high toxicity. Therefore, it is important to develop a safe and reliable treatment strategy that meets with high patient compliance. Wu et al. [128] developed an AAV-delivered muscone-induced transgene system (AAVMUSE) to treat non-alcoholic fatty liver disease (NAFLD) and allergic asthma. One injection of AAVMUSE, and then regular quantitative inhalation of musk ketone, can continuously induce the expression of proteins in situ, achieving the purpose of “smelling fragrance” to treat chronic diseases. The results showed that, in NAFLD model mice, AAVMUSE was able to precisely regulate the production of ΔhFGF21 after delivery to liver tissue via AAV2/9, and achieved treatment for up to 28 weeks.

### 6.3. Lung Cancer

Lung cancer is one of the most prevalent cancers and the leading cause of death of all tumors (18.4% of all cancers) [129]. With the development of gene technology and immune therapy, gene therapy for lung cancer has emerged as a promising method. Various specific genes can serve as targets for the inhalation of gene delivery, including siRNA against highly expressed oncogenes or normal mRNA for downregulated tumor suppressor genes. The common vectors employed in these therapies contain liposomes, polymers and other biodegradable formulations. For example, the nebulization of PEI carrying Wilm’s tumor gene 1 (WT1-siRNA) has shown promise in the treatment of melanoma lung cancer metastasis, which can effectively attenuate tumor progression [72]. Biocompatible formulations have gained increased attention. The inhalation of glycerol propoxylate triacrylate and spermine (GPT-SPE) carrying endoplasmic reticulum (ER)–Golgi intermediate compartment (ERGIC3) short hairpin RNA (shERGIC3) has been proven to suppress lung cancer tumorigenesis in K-Ras mutant mice [130].

## 7. Conclusions and Future Perspectives

Aerosol gene delivery has been widely explored. Notably, the inhalation of chemotherapy has preceded the inhalation of gene delivery. For example, one of the earliest clinical trials in inhalation involved the pulmonary administration of 5-fluorouracil (5-FU) for non-small-cell lung cancer (NSCLC) [131]. The results demonstrated reduced systemic side effects with a lower plasma concentration and increased accumulation at the lung tumor site. Nowadays, gene therapy has gained widespread popularity as a new treatment method for numerous diseases. However, the development of efficient and low cytotoxic carriers for gene delivery still requires further research. In addition to the vectors mentioned before, there are many other materials applied for delivering genes. There are typically two major gene delivery systems, viral vectors and non-viral vectors [132]. Although viral vectors offer superior transfection efficiency, they also face the challenges of immunogenicity, off-target effects, instability and infectivity [133]. A few clinical trials have experienced setbacks in viral gene therapy. Biomaterials have emerged as an alternative to viral vectors because they are more controllable, much safer and programmable [134]. Non-viral vectors encompass polymers, liposomes, peptide-modified polyplexes and other natural biocompatible vectors. In addition to these classic carriers that have been extensively explored for inhalation, other vectors such as metal nanoparticles and mesoporous silica have also been reported. For example, Ju et al. [135] designed a metal–phenolic capsule that allows for size regulation through the repeated film deposition of phenols. The formulation is proven to be biodegradable after intratracheal administration and shows prolonged retention in lungs for up to 24 h. This tunable capsule demonstrates good aerodynamic properties and loading capacities. The prospect of aerosol gene delivery seems to offer various opportunities for quite a few pulmonary diseases, ranging from lung cancer [136] to chronic diseases like CF. Nevertheless, it still faces the challenges of safety, biocompatibility, transfection efficiency and stability, which need to be addressed before the ultimate application. The main issue is that the carriers should remain intact under high shear force during aerosol inhalation. Although the inhalation of gene delivery can reduce systemic side effects compared to intravenous administration, it still cannot realize complete targeting to specific lung sites. For instance, the inhalation of nanoparticles for lung cancer is largely based on the enhanced permeation and retention effect (EPR). It relies more on the passive target. Some researchers have endeavored to modify the surface of nanoparticles to target specific receptors such as the luteinizing hormone-releasing hormone (LHRH) receptor, folate receptor, transferrin receptor and epidermal growth factor receptor (EGFR) [137]. These modified vectors can achieve better site-specific targeting.

Although a lot of research has been devoted to aerosol gene delivery, it still faces several challenges. First, finding a suitable delivery vector needs further exploration. Biosafety, the immune response and mutation caused by the viral vector are still potential problems [138]. Second, the improvement of the nebulizer device and dose control is required to ensure sufficient lung accumulation [33]. Otherwise, particles may be deposited in other sites instead of the deep lung. Third, the internalization of the aerosol inhalation of genes into target cells is key for achieving therapeutic effects. Many gene carrier particles will be phagocytized by lung macrophages before they can enter the target cells. What is more, the safety of aerosol inhalation in younger people, even infants, has not been validated and requires long-term evaluation.

In conclusion, aerosol gene delivery treatment is a promising future direction, but further research is needed to address the aforementioned concerns and ensure continuous improvement.

## Figures and Tables

**Figure 1 biomolecules-14-00904-f001:**
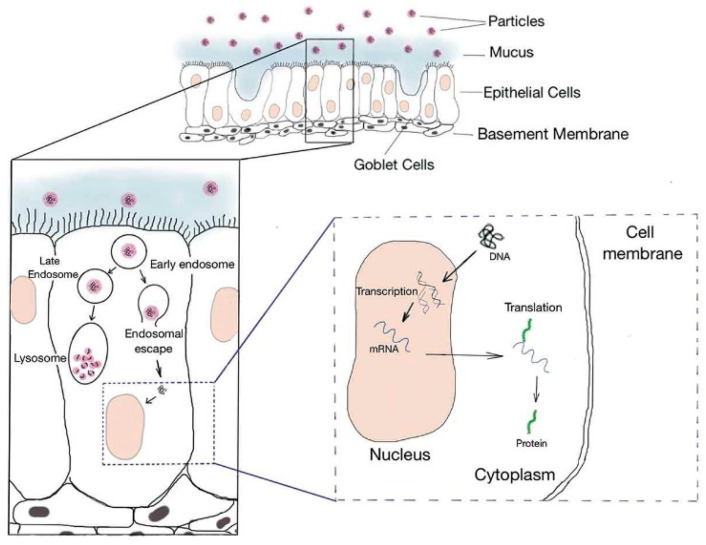
Illustration of the pathway of aerosol gene delivery [31]: 1. particles dissolve in the fluid and are transcribed into the cells; 2. they escape from the endosome or are degraded by lysosomes; 3. DNA gets into the nucleus and is transcribed into RNA, which in turn is translated into proteins.

**Figure 2 biomolecules-14-00904-f002:**
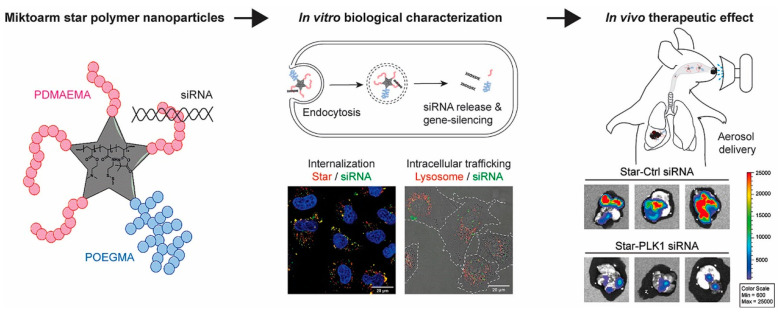
Illustration of the synthetic and in vitro and in vivo function of star-siRNA polymers [65].

**Figure 3 biomolecules-14-00904-f003:**
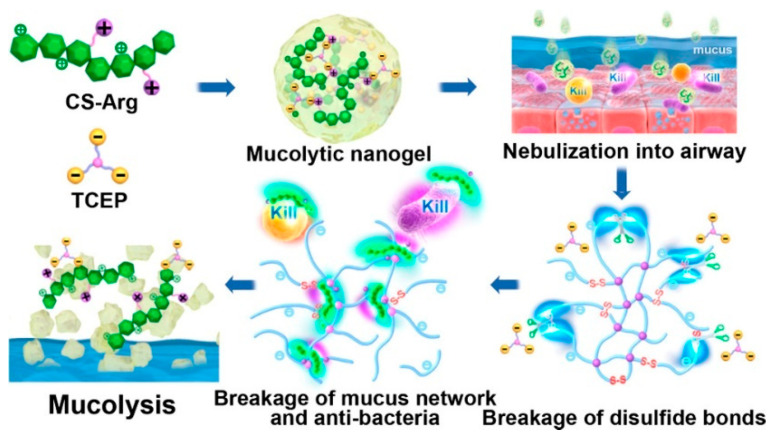
The formulation of CAT [95].

**Figure 4 biomolecules-14-00904-f004:**
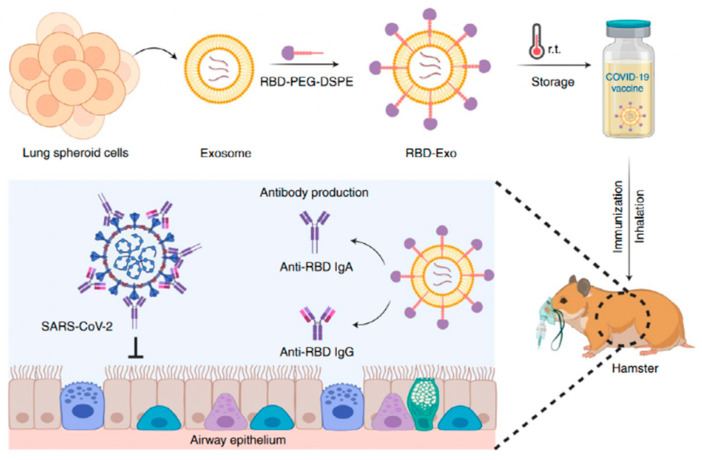
Preparation of RBD-LSC-Exo [100].

**Figure 5 biomolecules-14-00904-f005:**
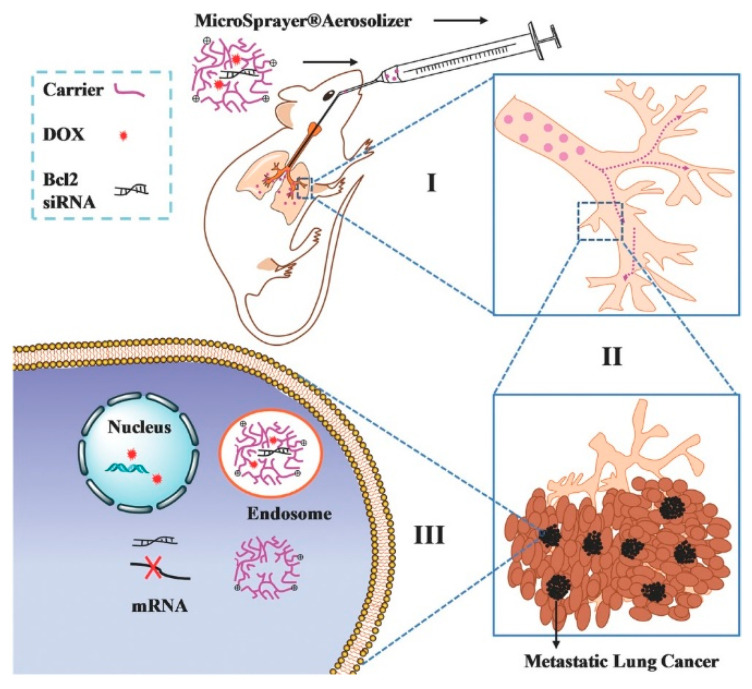
Intratracheal delivery of PEI-Bcl2-siRNA [111].

**Table 1 biomolecules-14-00904-t001:** Different gene carriers used in aerosol delivery.

Viral vectors	Lentivirus
	Adeno-associated virus vector (AAV)
	Adenovirus
Non-viral vectors	Liposomes
	Polymers (poly-ethylenimine (PEI), other polyplexes, synthetic esters)
	Peptide-based nanoparticles
	Natural biocompatible components (chitosan or exosome-based vectors)

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
