# Peer review of "Aerosol Inhalation of Gene Delivery Therapy for Pulmonary Diseases"

_biomolecules, 2024, doi:10.3390/biom14080904_

Round 1

Reviewer 1 Report

Comments and Suggestions for Authors

Huang et. al studied Aerosol inhalation of gene delivery therapy for pulmonary diseases. There is a need for extensive revision of this manuscript before it can be published.

1)      The author must explain the use of aerosolized gene therapy to treat lung diseases and aerosolized vaccines to prevent influenza and measles outbreaks.

2)      The biological functions of CRISPR/Cas9 systems may be hindered in their clinical translation due to the lack of progress in inhalation formulation and delivery – explain it.

3)      (a) List drugs that treat non-pulmonary diseases and target the systemic circulation (systemic drug delivery); (b) Provide nucleic acids that result in lasting expression of a gene construct or protein coding sequence in a group of cells (gene therapy); and (c) Explain vaccination without the need for needles (via aerosolized immunization).

4)      Explain of the role of aerosol therapy (a) Enhancing the effectiveness of drugs delivered systemically; (b) Creating gene therapy vectors that can effectively enter cells and deliver DNA material to the nucleus; (c) Enhancing the transportation of gene vectors and vaccines to infants.; and (d) Working on developing safe formulations for both acute and chronic treatments.

5)      Clinical trials offer the most trustworthy evaluation of intervention effects, including both efficacy and harm. However, unbiased assessment necessitates meticulous attention to design, conduct, analysis, and ethical considerations. Explain all these in your manuscript.

Go through this paper:

(a)   Success of nano-vaccines against COVID-19: a transformation in nanomedicine: Expert Review of Vaccines: Vol 21, No 12 (tandfonline.com)

Comments on the Quality of English Language

Moderate english editing required.

Reviewer 2 Report

Comments and Suggestions for Authors

I appreciate the thorough review and explanation of inhalational gene therapy. A nice addition would be a table summarizing the differenet technologies, vectors, etc. 

Suggest confirming factual statements related to any drug approval. For instance:

Line 547 – MRT5005 was not approved for treatment of CF by the FDA, as stated.

Comments on the Quality of English Language

line 99 – “participate”

line 101 – “build”

line 102 – “maintaining”

line 107 – “is transcribed”

line 112- “approach”

line 120 “characteristics”

line 170 – fix grammar

Line 217 “indicated that the efficiency”

line 218 delete “an” after maintaining.

Line 223  - “featured”

Line 471: “showed that after”

Line 479: “expression, among which siRNA..”

Line 502 – fix sentence

Line 535- consider revising “treatment for CF remains unidentified.”

Line 547 – MRT5005 was not approved for treatment of CF by the FDA

Line 555 – “Aguero et al designed”